# Preliminary Evidence of Good Safety Profile and Outcomes of Early Treatment with Tixagevimab/Cilgavimab Compared to Previously Employed Monoclonal Antibodies for COVID-19 in Immunocompromised Patients

**DOI:** 10.3390/biomedicines11061540

**Published:** 2023-05-26

**Authors:** Andrea Lombardi, Giulia Viero, Simone Villa, Simona Biscarini, Emanuele Palomba, Cecilia Azzarà, Nathalie Iannotti, Bianca Mariani, Camilla Genovese, Mara Tomasello, Anna Tonizzo, Marco Fava, Antonia Grazia Valzano, Letizia Corinna Morlacchi, Maria Francesca Donato, Giuseppe Castellano, Ramona Cassin, Maria Carrabba, Antonio Muscatello, Andrea Gori, Alessandra Bandera

**Affiliations:** 1Infectious Diseases Unit, Foundation IRCCS Ca’ Granda Ospedale Maggiore Policlinico, 20122 Milan, Italy; 2Centre for Multidisciplinary Research in Health Science (MACH), University of Milano, 20122 Milan, Italy; 3Department of Pathophysiology and Transplantation, University of Milan, 20122 Milan, Italy; 4Clinical Laboratory, Foundation IRCCS Ca’ Granda Ospedale Maggiore Policlinico, 20122 Milan, Italy; 5Respiratory Unit and Cystic Fibrosis Adult Center, Foundation IRCCS Ca’ Granda Ospedale Maggiore Policlinico, 20122 Milan, Italy; 6A.M. & A. Migliavacca Center for Liver Disease, Division of Gastroenterology and Hepatology, Foundation IRCCS Ca’ Granda Ospedale Maggiore Policlinico, 20122 Milan, Italy; 7Department of Nephrology, Dialysis, and Renal Transplantation, Foundation IRCCS Ca’ Granda Ospedale Maggiore Policlinico, 20122 Milan, Italy; 8Hematology Unit, Foundation IRCCS Ca’ Granda Ospedale Maggiore Policlinico, 20122 Milan, Italy; 9Department of Internal Medicine, Adult Primary Immunodeficiencies Centre, Foundation IRCCS Ca’ Granda Ospedale Maggiore Policlinico, 20122 Milan, Italy

**Keywords:** monoclonal antibodies, tixagevimab/cilgavimab, immunocompromised

## Abstract

Objectives: Monoclonal antibodies (mAbs) have proven to be a valuable tool against COVID-19, mostly among subjects with risk factors for progression to severe illness. Tixagevimab/cilgavimab (TIX/CIL), a combination of two Fc-modified human monoclonal antibodies, has been recently approved to be employed as early treatment. Methods: Two groups of immunocompromised patients exposed to different early treatments (i.e., TIX/CIL vs. other mAbs [casirivimab/imdevimab, bamlanivimab/etesevimab, sotrovimab]) were compared in terms of clinical outcomes (hospitalisation and mortality within 14 days from administration) and time to the negativity of nasal swabs. We used either Pearson’s chi-square or Fisher’s exact test for categorical variables, whereas the Wilcoxon rank–sum test was employed for continuous ones. Kaplan–Meier curves were produced to compare the time to nasopharyngeal swab negativity. Results: Early treatment with TIX/CIL was administered to 19 immunocompromised patients, while 89 patients received other mAbs. Most of them were solid organ transplant recipients or suffering from hematologic or solid malignancies. Overall, no significant difference was observed between the two groups regarding clinical outcomes. In the TIX/CIL group, one patient (1/19, 5.3%), who was admitted to the emergency room within the first 14 days from treatment and was hospitalised due to COVID-19 progression, died. Regarding the time to nasal swab negativity, no significant difference (*p* = 0.088) emerged. Conclusions: Early treatment of SARS-CoV-2 infection with TIX/CIL showed favourable outcomes in a small group of immunocompromised patients, reporting no significant difference compared to similar patients treated with other mAbs.

## 1. Introduction

Since the coronavirus disease 2019 (COVID-19) pandemic began, vaccination has represented the primary strategy to contain the diffusion of the virus. However, some subjects do not retain a competent immune system and, therefore, might not respond with a complete adaptive immune response. Thus, the concept of passive immunisation through the use of monoclonal antibodies has been deeply explored. These molecules have become part of the therapeutic strategy against SARS-CoV-2 with their innate capacity to protect against disease irrespective of one’s immune system status. Early treatment with monoclonal antibodies has proven to be a valuable tool against COVID-19, mostly among subjects with risk factors for progression to severe illness [1]. However, their use has been jeopardised by the emerging variants of concern, whose resistant strains might compromise their efficacy.

Tixagevimab/cilgavimab (TIX/CIL) is a combination of two Fc-modified human monoclonal antibodies (mAbs) which bind simultaneously to distinct non-overlapping epitopes of Spike protein receptor binding domain. It was initially developed to be employed as a primary prophylaxis tool among those unable to receive the vaccination or with conditions impairing the response to immunisation programs [2,3]. In the summer of 2022, based on the positive results of two phases 3 trials [4,5], its indications were also expanded to early treatment to prevent progression to more severe COVID-related manifestations and outcomes. Among the patients who may benefit more from early mAbs treatments are immunocompromised individuals (e.g., solid organ transplant recipients, those receiving immunosuppressive drugs for autoimmune conditions, and those with primary immunodeficiencies), a population which is usually excluded or under-represented in registration studies.

We have previously observed minimal adverse drug reactions and favourable outcomes among immunocompromised patients receiving early treatment with the mAbs casirivimab/imdevimab, sotrovimab, or bamlanivimab/etesevimab [6]. This study aims to assess clinical outcomes and time to nasal swab negativity in a cohort of immunocompromised patients treated with TIX/CIL.

## 2. Methods

The study included immunocompromised patients [(i) history of any connective tissue disease, autoimmune disease, or primary immunodeficiency; (ii) history of an active solid or hematologic tumour; (iii) neutropenia due to haematological cancer; (iv) diagnosis of human immunodeficiency virus (HIV) infection or acquired immunodeficiency syndrome (AIDS); (v) history of splenectomy, solid organ transplantation (SOT), and/or hematopoietic stem cell transplantation (HSCT); or (vi) ongoing treatment with steroids (for at least four weeks), chemotherapy, and/or immunosuppressive agents], with COVID-19 diagnosis, evaluated at the outpatient clinic or hospitalised for a non-COVID-19-related reason in the ward of the Infectious Diseases Unit, IRCCS Ospedale Maggiore Policlinico, Milano, Italy, from 28 August to 15 October 2022, who received early treatment with TIX/CIL. This group was compared to subjects who had received other mAbs (casirivimab/imdevimab, bamlanivimab/etesevimab, sotrovimab) between 25 November 2021 and 25 January 2022, as previously published [6].

We compared clinical outcomes (i.e., hospitalisation and mortality within 14 days from administration) and time to the negativity of nasal swabs. Categorical variables were compared by using either Pearson’s chi-square or Fisher’s exact test, whereas the Wilcoxon rank–sum test was employed for continuous variables. Kaplan–Meier curves were produced to compare the time to nasopharyngeal swab negativity.

The study was conducted in accordance with the Declaration of Helsinki and approved by the Institutional Review Board of IRCCS Fondazione Ca’ Granda Ospedale Maggiore Policlinico (protocol code Milano Area 2, #328_2022bis, 26 April 2022).

## 3. Results

Early treatment with TIX/CIL was administered to 19 immunocompromised patients, whereas 89 individuals were treated with other mAbs. The majority of patients included in the TIX/CIL cohort were SOT or individuals suffering from hematologic or solid malignancies. TIX/CIL treatment was administered on average 5(±5) days after symptom occurrence. Table 1 summarises the demographic and clinical characteristics of the enrolled patients. Table 2 reports clinical outcomes compared between the two groups.

Overall, no significant difference was observed. In the TIX/CIL cohort, one patient (1/19, 5.3%), who was admitted to the emergency room within the first 14 days from treatment and was hospitalised due to COVID-19 progression, died. Regarding the time to nasal swab negativity, no significant difference (*p* = 0.088) emerged between the two groups, with 36/89 (40.4%) and 5/19 (26.3%) of patients being negative at 14 days since treatment administration in the mAbs and TIX/CIL group, respectively (Figure 1).

Appendix A describes signs and symptoms displayed by enrolled patients at the time of treatment evaluation; the only difference was a lower frequency of fever among the TIX/CIL patients. Appendix A provides details about the different mAbs administered and the vaccine doses received by the enrolled patients. Overall, the mAb most frequently administered was sotrovimab, whereas patients in the TIX/CIL group received more vaccine doses than those in the mAb cohort.

## 4. Discussion

In our study, early treatment of SARS-CoV-2 infection with TIX/CIL showed favourable outcomes in a small group of immunocompromised patients, reporting no significant difference compared to subjects with comparable health conditions treated with other mAbs. Likewise, the time to the negativity of nasal swabs was not different among the treatments.

Our findings were obtained in Italy in August–October 2022, after the approval of TIX/CIL as an early COVID-19 treatment by the Agenzia Italiana del Farmaco [7]. In this timeframe, the SARS-CoV-2 variants of concern (VOCs) predominant in the Italian territory were Omicron BA.4 and BA.5 [8]; thus, our data can be applied to a setting where these VOCs, or others with susceptibility to TIX/CIL combination, are those most frequently responsible for the infection.

Clinically significant protection against progression to severe COVID-19 or death has been demonstrated for TIX/CIL early treatment in significant phase 3 registration studies [4,5]. While evidence is accumulating showing the efficacy of TIX/CIL primary prophylaxis among immunocompromised patients [9,10,11], less is known about the specific impact of TIX/CIL early treatment among this group of patients. Overall, in the ACTIV-3/TICO and TACKLE studies, the numbers of immunocompromised patients were 57 (8%) and 22 (5%), respectively [4,5]. A French group recently led a retrospective single-centre study to analyse clinical outcomes in a small cohort of patients with haematologic malignancies treated with tixagevimab-cilgavimab for infection with the Omicron SARS-CoV-2 variant (subvariants BA.1 and BA.2). Despite the small sample of patients; it was the first study to describe outcomes after curative treatment with tixagevimab-cilgavimab for infection caused by the Omicron variant. The results showed how asymptomatic and paucisymptomatic patients, despite presenting high-risk factors for progression, did not develop a severe disease [12]. Along with this evidence, a Greek group recently published a review on the TIX/CIL use in the prevention and early treatment, confirming the limited experience of therapeutic administration among immunocompromised subjects [13]. Our study is therefore pioneering in providing preliminary evidence for this vulnerable group of people, employing other mAbs as comparators and not placebo, thus reflecting more accurately the real-life experience.

Unfortunately, new variants and descendant subvariants keep emerging, contributing to the so-called “variant soup”. In this rapidly evolving setting, most of the used monoclonal antibodies have lost their efficacy: sotrovimab, casirivimab/imdevimab and bamlanivimab/etesevimab were effective against the previous variants and the first Omicron descending subvariants (B.1.1.529/BA.1 and BA.1.1), but have been described as inactive against the new ones (BA.2, BA.4, BA.5). Recently published in vitro data has suggested how emerging Omicron sub-lineages are resistant to most (i.e., BA.4.6, BA.2.75.2, and BJ.1) or all (BQ.1.1) mAbs used in routine practices, including TIX/CIL [11]. Along with these, two novel subvariants (XBB, XBB.1) are the most resistant to date, with a described in vitro inactivation of all clinical monoclonal antibodies in use [14]. As infections due to VOCs BQ.1/1.1 are skyrocketing in Western countries, including Italy, our data might soon become less relevant because of the rapidly evolving epidemiology [8].

Our study has some inherent limitations related to its retrospective design. Particularly, TIX/CIL treatment has been compared with a historical group of patients treated with other mAbs, with different SARS-CoV-2 VOCs representing the dominant strain at the time and with a population who received fewer vaccine doses and experienced fewer past SARS-CoV-2 infections. Nonetheless, considering the similarity of the patients included in the two study groups, and the impossibility for ethical reasons to compare TIX/CIL with mAbs with known inefficacy against Omicron VOC, we believe our results are still interesting. Another theoretical limit is the follow-up time for clinical outcomes being restricted to 14 days since treatment administration, which may have reduced the detection of long-term outcomes due to COVID-19.

Overall, TIX/CIL early treatment has demonstrated favourable outcomes among immunocompromised patients, supporting its employment in this population, which usually does not have access to other therapies because of drug–drug interactions (i.e., nirmatrelvir/ritonavir and tacrolimus in SOT) or comorbidities (i.e., nirmatrelvir/ritonavir or molnupiravir among patients with estimated glomerular filtration rate <30 mL/min). TIX/CIL should be offered as early treatment until the evolution of circulating VOCs leads to its ineffectiveness.

Evidence on the effectiveness of TIX/CIL treatment in clinical practice is limited, specifically among fragile subjects, who have been poorly represented in major randomised controlled trials but may benefit the most from these approaches. There is, therefore, an urgent need to shed light on the safety, efficacy, and long-term outcomes of early treatment with TIX/CIL among this peculiar population, especially in the context of SARS-CoV-2 VOCs BQ.1/1.1 diffusion.

## Figures and Tables

**Figure 1 biomedicines-11-01540-f001:**
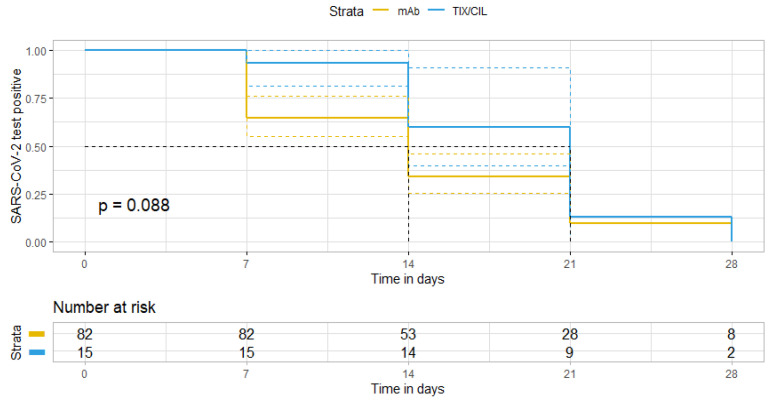
Kaplan-Meier curve showing time from treatment initiation and SARS-CoV-2 nasal swab negativity.

**Table 1 biomedicines-11-01540-t001:** Demographic and clinical characteristics of patients undergoing early treatment with TIX/CIL and other mAbs.

Characteristic	Overall, N = 108	mAbs, N = 89	TIX/CIL, N = 19	*p*-Value
**Age**				0.003
20–64 years	80 (74%)	61 (69%)	19 (100%)	
65+ years	28 (26%)	28 (31%)	0 (0%)	
**Sex**				0.377
M	61 (56%)	52 (58%)	9 (47%)	
F	47 (44%)	37 (42%)	10 (53%)	
**Ethnic group**				0.433
Caucasian	82 (76%)	82 (92%)	18 (95%)	
African	21 (19%)	3 (3.4%)	0 (0%)	
Asian	3 (2.8%)	3 (3.4%)	0 (0%)	
Hispanic	2 (1.9%)	1 (1.1%)	1 (5.3%)	
**BMI**	24.0 (21.2, 26.0)	24.0 (21.0, 26.0)	25.0 (22.0, 30.0)	0.154
**Hypertension**	40 (37%)	29 (33%)	11 (58%)	0.038
**Potus**	2 (2.1%)	2 (2.3%)	0 (0%)	>0.999
**Smoke**				0.159
Never	65 (66%)	54 (62%)	11 (92%)	
Former smoker	18 (18%)	17 (20%)	1 (8.3%)	
Active smoker	16 (16%)	16 (18%)	0 (0%)	
**Previous SARS-CoV-2 infection**	11 (10%)	5 (5.7%)	6 (32%)	0.004
**Connective tissue disease**	12 (11%)	11 (12%)	1 (5.3%)	0.688
**Solid tumour**				>0.999
None	100 (93%)	82 (92%)	18 (95%)	
Local	6 (5.6%)	5 (5.6%)	1 (5.3%)	
Metastatic	2 (1.9%)	2 (2.2%)	0 (0%)	
**Leukaemia**	7 (6.5%)	5 (5.6%)	2 (11%)	0.604
**Lymphoma**	12 (11%)	10 (11%)	2 (11%)	>0.999
**AIDS**	0 (0%)	0 (0%)	0 (0%)	>0.999
**Splenectomy**	2 (1.9%)	2 (2.2%)	0 (0%)	>0.999
**Neutropenia**	3 (2.8%)	1 (1.1%)	2 (11%)	0.079
**Primary immunodeficiency**	23 (21%)	21 (24%)	2 (11%)	0.354
**Autoimmune disease**	14 (13%)	13 (15%)	1 (5.3%)	0.456
**Bone marrow transplant**				>0.999
No	104 (96%)	85 (96%)	19 (100%)	
Autologous	4 (3.7%)	4 (4.5%)	0 (0%)	
Allogenic	0 (0%)	0 (0%)	0 (0%)	
**Solid organ transplant**				0.002
No	58 (55%)	53 (60%)	5 (28%)	
Kidney	26 (25%)	22 (25%)	4 (22%)	
Liver	14 (13%)	10 (11%)	4 (22%)	
Lungs	8 (7.5%)	3 (3.4%)	5 (28%)	
Other(s)	0 (0%)	0 (0%)	0 (0%)	
**HIV infection**	2 (1.9%)	2 (2.2%)	0 (0%)	>0.999
**Long-term steroid**				0.024
No	53 (49%)	48 (54%)	5 (26%)	
<20 mg/die	49 (45%)	38 (43%)	11 (58%)	
≥20 mg/die	6 (5.6%)	3 (3.4%)	3 (16%)	
**Biological immunosuppressor**				0.019
Anti TNF-alfa	1 (6.7%)	1 (7.7%)	0 (0%)	
Anti IL6	1 (6.7%)	1 (7.7%)	0 (0%)	
Anti IL1	0 (0%)	0 (0%)	0 (0%)	
Anti IL17a	0 (0%)	0 (0%)	0 (0%)	
Anti CD20	3 (20%)	3 (23%)	0 (0%)	
TK inhibitors	2 (13%)	0 (0%)	2 (100%)	
Anti CD52	0 (0%)	0 (0%)	0 (0%)	
Other(s)	8 (53%)	8 (62%)	0 (0%)	
**Chemotherapy**	7 (6.5%)	5 (5.6%)	2 (11%)	0.604
**Anti-rejection therapy**	51 (47%)	38 (43%)	13 (68%)	0.041

**Table 2 biomedicines-11-01540-t002:** COVID-19-related clinical outcomes.

COVID-19 Outcomes	Overall, N = 108	mAbs, N = 89	TIX/CIL, N = 19	*p*-Value
**Hospital admission within 14 days from infusion**	8 (7.4%)	7 (7.9%)	1 (5.3%)	>0.999
of which related to COVID-19	5 (4.6%)	4 (4.5%)	1 (5.3%)	>0.999
**Emergency department admission within 14 days from infusion**	4 (3.7%)	3 (3.4%)	1 (5.3%)	0.544
**ICU admission within 14 days from infusion**	0 (0%)	0 (0%)	0 (0%)	
**Death within 14 days from infusion**	2 (1.9%)	1 (1.1%)	1 (5.3%)	0.322
of which related to COVID-19	2 (1.9%)	1 (1%)	1 (5.3%)	0.322

ICU: intensive care unit.

## Data Availability

Data will be provided on reasonable request.

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
