# Peer review of "Preliminary Evidence of Good Safety Profile and Outcomes of Early Treatment with Tixagevimab/Cilgavimab Compared to Previously Employed Monoclonal Antibodies for COVID-19 in Immunocompromised Patients"

_biomedicines, 2023, doi:10.3390/biomedicines11061540_

Round 1

Reviewer 1 Report

The authors do not reference an article reviewing “Tixagevimab/Cilgavimab in SARS-CoV-2 Prophylaxis and Therapy: A Comprehensive Review of Clinical Experience” (Viruses 2023, 15, 118. https://doi.org/10.3390/v15010118) and in which the conclusions already pointed to what is presented here.

In this way, I consider that it does not add anything new to the recently published works on the clinical use of the two antibodies in the treatment of cases of immunocompromised patients with covid.

In addition, there is an inconsistency in the way the bibliographic references are presented:

Lines 147 and 157

Why authors use DOI (https://doi.org/ 10.1038/s41409-022-01894-1) (doi: 10.1016/j.cell.2022.12.018) and did not present these references in the reference section?

Author Response

Milano, 21 May 2023

Ref: biomedicines-2197295

Ms Alexandra Negura
Assistant Editor, MDPI Romania

Dear Ms Alexandra Negura,

Thank you for the evaluation provided of our manuscript. Please find below our point-by-point answers to the reviewers' comments. We have also discussed as suggested by the academic editor, the manuscript "Tixagevimab/Cilgavimab in SARS-CoV-2 Prophylaxis and Therapy: A Comprehensive Review of Clinical Experience" (Viruses 2023, 15, 118. https://doi.org/10.3390/v15010118).

We do look forward to a favourable consideration and remain.

On behalf of all the authors,

Andrea Lombardi, MD

Department of Pathophysiology and Transplantation, University of Milano

Infectious Diseases Unit, IRCCS Ca' Granda Ospedale Maggiore Policlinico Foundation

Via Francesco Sforza 35, 20122, Milan, Italy

Tel: +390255034767 Skype: andrea.lombardi89

Reviewer 1

The authors do not reference an article reviewing "Tixagevimab/Cilgavimab in SARS-CoV-2 Prophylaxis and Therapy: A Comprehensive Review of Clinical Experience" (Viruses 2023, 15, 118. https://doi.org/10.3390/v15010118) and in which the conclusions already pointed to what is presented here.

In this way, I consider that it does not add anything new to the recently published works on the clinical use of the two antibodies in the treatment of cases of immunocompromised patients with covid.

In addition, there is an inconsistency in the way the bibliographic references are presented: Lines 147 and 157. Why authors use DOI (https://doi.org/ 10.1038/s41409-022-01894-1) (doi: 10.1016/j.cell.2022.12.018) and did not present these references in the reference section?

Re: We thank the reviewer for his evaluation. We have added the reference above in the discussion section, please see lines 203-206. Unfortunately, the article to which the reviewer is referring is an excellent summary of the literature but does not provide a metanalysis of the mAbs efficacy, especially in the subgroup of SOT patients. Therefore, our data can be a helpful addition to the bulk of literature available that can be included in a future systematic review and metanalysis.

We have also corrected the typo regarding DOI.

Reviewer 2

This paper by Lombardi et al. reports the efficacy of a combination monoclonal against SARS-COVID. Recently formulated, this antibody mix has been dubbed TIX/CIL, and clinically tested before with some success. This paper examines the results in a cohort of patients in Italy, and confirms its protective ability against the SARS-CoV strains / variants, circulating on the test population.

The study is straightforward, logical, and detailed but also short and focused. My only comment is regarding the future of TIX/CIL in the clinic. The authors have shown that its effectivity is virtually identical to that of the other monoclonals that are already in use. In other words, it is neither better nor worse than them. So, what is its unique benefit that the others do not have? Is it just another monoclonal arsenal (against a different epitope of the spike protein) that may "potentially" be more effective against a future variant? This aspect could be stated more clearly.

Re: We thank the reviewer for his excellent evaluation. We agree with him regarding the problematic future of the drug in the clinical practice because of the continuous viral evolution. We have stated that in lines 210-214.

Reviewer 2 Report

This paper by Lombardi et al. reports the efficacy of a combination monoclonal against SARS-COVID. Recently formulated, this antibody mix has been dubbed TIX/CIL, and clinically tested before with some success. This paper examines the results in a cohort of patients in Italy, and confirms its protective ability against the SARS-CoV strains / variants, circulating on the test population.

The study is straightforward, logical, and detailed but also short and focused. My only comment is regarding the future of TIX/CIL in the clinic. The authors have shown that its effectivity is virtually identical to that of the other monoclonals that are already in use. In other words, it is neither better nor worse than them. So, what is its unique benefit that the others do not have? Is it just another monoclonal arsenal (against a different epitope of the spike protein) that may "potentially" be more effective against a future variant? This aspect could be stated more clearly.

Author Response

(The authors gave the same response as above.)

Round 2

Reviewer 1 Report

The authors improved the aspects related to the bibliography and gave an explanation for the relevance of the study they are presenting, in this way the communication can be considered for publication.